# Early Forms of *α*-Synuclein Pathology Are Associated with Neuronal Complex I Deficiency in the Substantia Nigra of Individuals with Parkinson’s Disease

**DOI:** 10.3390/biom12060747

**Published:** 2022-05-25

**Authors:** Irene Hana Flønes, Harald Nyland, Dagny-Ann Sandnes, Guido Werner Alves, Ole-Bjørn Tysnes, Charalampos Tzoulis

**Affiliations:** 1Neuro-SysMed, Department of Neurology, Haukeland University Hospital, 5021 Bergen, Norway; irene.flones@uib.no (I.H.F.); harald.nyland@student.uib.no (H.N.); da.sandnes@gmail.com (D.-A.S.); ole-bjorn.tysnes@helse-bergen.no (O.-B.T.); 2Department of Clinical Medicine, University of Bergen, Pb 7804, 5020 Bergen, Norway; 3K.G. Jebsen Center for Translational Research in Parkinson’s Disease, University of Bergen, Pb 7804, 5020 Bergen, Norway; 4The Norwegian Centre for Movement Disorders and Department of Neurology, Stavanger University Hospital, Pb 8100, 4068 Stavanger, Norway; guido.werner.alves@sus.no; 5Department of Mathematics and Natural Sciences, University of Stavanger, 4062 Stavanger, Norway

**Keywords:** OXPHOS, mitochondria, synucleinopathy, α-synuclein, Lewy-pathology, respiratory chain

## Abstract

Idiopathic Parkinson’s disease (iPD) is characterized by degeneration of the dopaminergic substantia nigra pars compacta (SNc), typically in the presence of Lewy pathology (LP) and mitochondrial respiratory complex I (CI) deficiency. LP is driven by α-synuclein aggregation, morphologically evolving from early punctate inclusions to Lewy bodies (LBs). The relationship between α-synuclein aggregation and CI deficiency in iPD is poorly understood. While studies in models suggest they are causally linked, observations in human SNc show that LBs preferentially occur in CI intact neurons. Since LBs are end-results of α-synuclein aggregation, we hypothesized that the relationship between LP and CI deficiency may be better reflected in neurons with early-stage α-synuclein pathology. Using quadruple immunofluorescence in SNc tissue from eight iPD subjects, we assessed the relationship between neuronal CI or CIV deficiency
and early or late forms of LP. In agreement with previous findings, we did not observe CI-negative neurons with late LP. In contrast, early LP showed a significant predilection for CI-negative neurons (*p* = 6.3 × 10^−5^). CIV deficiency was not associated with LP. Our findings indicate that early α-syn aggregation is associated with CI deficiency in iPD, and suggest a double-hit mechanism, where neurons exhibiting both these pathologies are selectively lost.

## 1. Introduction

Parkinson’s disease (PD) is the most common neurodegenerative synucleinopathy. Its prevalence has shown a dramatic increase during the last three decades and it is considered one of the fastest growing neurological disorders worldwide [1,2]. Monogenic forms of PD are generally rare, accounting for less than 5% of all cases in most populations, and the vast majority of affected individuals have idiopathic PD (iPD) [1].

Hallmark pathological features of iPD are the loss of the dopaminergic neurons of the substantia nigra pars compacta (SNc) [3,4], and a region- and cell-specific occurrence of Lewy pathology (LP), of which aggregated α-synuclein (α-syn) is a major constituent [5]. The aggregation of α-syn has been shown to be a highly dynamic process, proposed to comprise several stages from early misfolding and oligomerization of α-syn to mature Lewy body (LB) formation [6]. Histologically, this evolution is believed to be reflected by different morphologies of aggregated α-syn, which in the SNc range from diffuse punctate inclusions, in early LP, to formed inclusions, such as pale bodies (PBs) and finally mature LBs [6,7]. Another consistently encountered feature of iPD is the presence of mitochondrial pathology in dopaminergic neurons of the SNc, including impaired mitochondrial DNA (mtDNA) maintenance [8,9,10], and quantitative and/or functional deficiencies of the mitochondrial respiratory chain (MRC), mostly affecting complex I (CI) [11].

A growing body of evidence from cell and animal studies suggests that LP and mitochondrial dysfunction may be causally linked in iPD and may act synergistically in driving neuronal dysfunction and death. However, the dynamics of the relationship between these two pathologies remain poorly understood. Experiments in mammalian cells show that α-syn aggregation inhibits CI function, and leads to loss of mitochondrial membrane potential, and excess generation of reactive oxygen species (ROS) [12,13,14,15,16]. Furthermore, α-syn knockout mice are resistant to Parkinsonism-inducing neurotoxins inhibiting CI [17], whereas mice overexpressing α-syn show increased vulnerability to the same toxins [18].

If α-syn aggregation and MRC deficiency are indeed causally linked in iPD, one would expect the two pathology states to correlate across neurons. However, immunohistochemical studies in the SNc of individuals with iPD reveal the opposite pattern, namely, LBs and PBs are more frequently encountered in CI-positive than in CI-negative dopaminergic neurons [19,20]. Conversely, dopaminergic neurons with LP show higher intensities of CI and complex IV (CIV) [19]. A proposed explanation for this inverse association is that, if co-occurrence of MRC deficiency and LP are synergistically deleterious, neurons that suffer a “double hit” would be preferentially lost and, therefore, less frequently observed. Thus, since formed PBs and LBs are end results of the α-syn aggregation process, the relationship between α-syn pathology and MRC deficiency may be better reflected in neurons with early-stage α-syn aggregation.

Here, we attempt to address this pertinent question by investigating the relationship between proposed histological signs of early or late LP and neuronal CI or CIV deficiency in the SNc of individuals with iPD. In agreement with previous findings [19,20], LBs occur preferentially in CI-positive neurons. In contrast, early α-syn aggregation shows a highly significant predilection for CI-negative over CI-positive neurons.

## 2. Materials and Methods

### 2.1. Subject Cohort and Tissue Samples

Brain tissue was obtained from eight individuals with clinicopathologically validated iPD from the Norwegian ParkWest study, a prospective population-based cohort of PD, systematically followed since 2004 [21], and from five neurologically healthy controls from our in-house brain bank. Known causes of monogenic PD had been excluded in the cases as previously described [22]. Brains were collected at autopsy and processed as previously described [10]. For the neuropathological evaluation, LP and neurofibrillary pathology were staged according to the Braak criteria [23,24] and β-amyloid phases were evaluated according to the Thal criteria [25]. Subject demographics, pathology scores, and clinical data are summarized in Table 1. There was no statistically significant difference in sex distribution, age of death, post-mortem interval (PMI), Braak stage (neurofibrillary pathology), or Thal phase between iPD and controls (not shown).

### 2.2. Immunohistochemistry of Aggregated α-syn

Immunohistochemistry (IHC) was carried out on formalin-fixed, paraffin-embedded SNc from all included individuals. The 3-μm-thick sections were stained with primary antibodies against α-syn: clone 5G4: Millipore, MABN38, dilution: 1:16,000; clone KM51: Leica, NCL-L-ASYN, dilution 1:50. The clone 5G4 antibody recognizes amino acids 46–53 and shows a high immunoreactivity to all forms of ß-sheet rich α-syn aggregates and a lower affinity towards α-syn monomers [26,27,28], whereas the clone KM51 antibody recognizes full length α-syn. IHC with the 5G4-antibody was carried out using the intelliPATH FLX Automated Slide Stainer (Biocare, Histolab Products AB, Askim, Sweden). Deparaffinization and antigen retrieval was performed in the low pH EnVision FLEX Target Retrieval Solution at 98 °C for 24 min in the DAKO PT link (Agilent DGG Norge AS, Oslo, Norway). To optimize visualization, tissues were treated in concentrated formic acid for 1 min following antigen retrieval. The antibody was diluted in DaVinci Green Diluent (Biocare, PD900) and incubated for 1 h at room temperature. MACH4 Universal HRP-polymer (Biocare M4U534) and DAB chromogen kit (Biocare DB801) were used for visualization. Tacha’s Automated Hematoxylin (Biocare, NM-HEM) was used for visualization of nuclei. Sections were washed in the TBS Automation Wash buffer (TWB945M, Biocare). IHC against α-syn KM51 was carried out at our local Department of Pathology in an automated stainer (Ventana Benchmark Ultra Stainer), and visualization performed using Ventana 760-500 Ultra View Universal DAB detection kit. The antibody was diluted in the Ventana Antibody Diluent (Ventana 251-018).

### 2.3. Fluorescence Immunohistochemistry

Quadruple fluorescence IHC with primary antibodies against the mitochondrial outer membrane protein VDAC1 (Abcam, Cambridge, UK, ab14734, dilution 1:100, α-syn 5G4 (Millipore, Merck Life Sciences AS, Oslo, Norway, MABN389, dilution 1:1000), mitochondrial CI (NDUFB10, Abcam, ab196019, dilution 1:100), or mitochondrial CIV (MTCOI, Invitrogen, Thermo Fisher Scientific Norge, Oslo, Norway, 459,600, dilution: 1:100) was carried out on 3-μm-thick sections of formalin-fixed, paraffin-embedded brain tissue from the SNc of individuals with iPD. Deparaffinization and antigen retrieval was performed using the low pH EnVision FLEX Target Retrieval Solution at 98 °C for 24 min in the PT link (Agilent DGG Norge AS, Oslo, Norway). To optimize α-syn visualization, tissues were treated in concentrated formic acid for 1 min following antigen retrieval. Permeabilization was performed using 1% Triton X-100 in Tris-buffered saline (TBS) for 15 min, blocking was performed at room temperature for 1 h in a TBS blocking buffer containing 3% normalized goat serum (Abcam, Ab138478) and 0.1% Triton X-100 (TBST). Primary and secondary antibodies were diluted in blocking buffer and incubated for 1 h at room temperature. Species- and IgG-subtype specific secondary antibodies were available from Thermo Fisher Scientific (Goat anti-Rabbit IgG Alexa Flour 488, A11008, Goat anti-Mouse IgG1 Alexa Fluor 594, A21125, Goat anti-Mouse IgG2b Alexa Flour 647, A21241, Goat-anti-mouse IgG2a Alexa Flour 488, A21131, all diluted at 1:200). Nuclei were visualized with DAPI diluted in TBST. Autofluorescence was quenched by incubation in Sudan Black B (Sigma-Aldrich, St. Louis, MO, USA, 199664-25G) diluted in 70% EtOH solution for 10 min. Sections were washed in TBST or TBS. Prolong Diamond Antifade mounting medium was used for mounting (P36961, Thermo Fisher Scientific, Thermo Fisher Scientific Norge, Oslo, Norway).

### 2.4. Image Analysis

Sections stained against the 5G4 and KM51 antibodies and visualized with DAB-chromogen were evaluated visually, and positive immunoreactions were compared between the two antibodies. Neuromelanin containing neurons of the SNc were identified based on their morphology and their topological relationship to the third cranial nerve and the red nucleus. In cases where the topography was not evident, both morphology and immunoreactivity against α-syn was used to identify the SNc. Images were taken at 40× magnification using the Leica DM 3000 LED microscope from Leica (Orthomedics AS, Lysaker, Norway).

Ten images from each fluorescently labeled section were taken at 40× magnification using a Leica DM 3000 LED microscope, equipped with a Leica ICC50 W camera and filter cubes detecting wavelengths at 405 nm (DAPI), 488 nm (NDUFB10 or MTCOI), 594 nm (VDAC1), and 647 nm (a-syn). Exposure times were set for each channel to minimize background, based on a negative control lacking primary antibody, and to avoid oversaturation. The same exposure times per channel were maintained for all samples. All neurons with VDAC1-positive immunoreaction (i.e., containing mitochondria) were classified according to the presence of CI (negative or positive) and α-syn staining by visual inspection. Two patterns of neuronal α-syn pathology were recognized: (1) LBs or PBs consistent with mature LP and (2) punctate inclusions consistent with early stages of LP [6,7]. Neurons exhibiting both patterns were included only in the first category. Image analysis was carried out using the Leica Application Suite X software version(v).17.0.0 from Leica-Microsystems GmbH (Wetzlar, Germany). Independent evaluation of the immunofluorescence images was performed by two researchers (IHF and HN). Figures were created in Fiji v.2.3.0 (National Institutes of Health and the Laboratory for Optical and Computational Instrumentation, University of Wisconsin, Madison, WI, USA), and CorelDraw Standard 2020 v.22.0.0 (Corel Corporation, Ottawa, ON, Canada).

### 2.5. Statistical Analysis

Subject demographics, pathology scores, and clinical data are given in Table 1. Data were tested for normality and equality of variances, using the Shapiro–Wilk test and Levene’s test. Between group-comparisons were performed using the Mann–Whitney U-test, due to data non-normality. Comparisons of grouped data were performed using Fisher’s exact test. Statistical analyses were carried out using SPSS v27.0.1.0. (IBM, New York, NY, USA) The Fisher–Freeman–Halton Exact Test was performed using the crosstab function in SPSS with CI or CIV state (positive/negative) as the dependent variable and α-syn inclusion (LB or PB/punctate inclusions/no inclusion) in each MRC complex as the independent variable. Inter-observer reliability was assessed by the intraclass correlation (ICC) coefficient or Cohen’s Kappa of each variable (i.e., number of observed neurons/CI state/CIV state/α-syn state). Graphic illustrations were created in GraphPad Prism v.9.3.1 (350), GraphPad Sofware, LCC, San Diego, CA, USA).

## 3. Results

### 3.1. The 5G4 Antibody Detects Numerous Intraneuronal and Glial Immunopositive Structures

As previously reported [27,28], the 5G4 antibody provided positive α-syn immunostaining in the absence of background staining or immunoreactivity of normal presynaptic structures. The 5G4 antibody identified a higher number of immunopositive punctate inclusions, extra-neuronal threads and grains, compared to the KM51 antibody, in the iPD cases (Figure 1). In addition, the 5G4 antibody detected positive inclusions in glia cells that were not observed in consecutive sections stained with the KM51 antibody. Neither antibody detected immunopositive staining in any of the controls, indicating that both antibodies are specific to pathogenic forms of α-syn.

### 3.2. Early α-syn Pathology Has a Predilection for CI-Negative Dopaminergic SNc Neurons

A total of 194 (observer 1) and 212 (observer 2) single dopaminergic neurons from the SNc of eight individuals were evaluated (Appendix A). One to two iPD individuals, depending on the observer, were deemed not to have CI-negative neurons in any of the assessed images, while the rest of the cases showed variable proportions of CI-negative dopaminergic neurons (Appendix A).

There was excellent inter-observer agreement [29] with regard to total number of observed neurons (ICC coefficient = 0.86, 95% C.I. (0.48, 0.97), *p* = 0.001), CI states (κ = 0.86, 95% C.I. (0.72, 0.99), *p* < 0.001), and α-syn states (κ = 0.86, 95% C.I. (0.78, 0.94), *p* < 0.001). Given this high concordance, we chose to focus on the data from observer-1, who was most experienced.

There was a significant difference in the distribution of LP between CI-positive and CI-negative neurons (Fisher–Freeman–Halton Exact Test: *p* = 9.0 × 10^−4^, 99% C.I. (1.3 × 10^−4^, 0.002)). Consistent with our previous report [19,20], we found no CI-negative neurons with LBs or PBs. However, this did not reach statistical significance when compared to LBs/PBs in CI-positive neurons in the post-hoc analysis (CI-positive: 23/176, 14%; CI-negative: 0/18, 0% *p* = 0.11). In contrast, punctate inclusions showed a highly significant predilection for CI-negative neurons (CI-positive: 28/176, 16 %; CI-negative: 10/18, 56%; *p* = 6.3 × 10^−5^). Conversely, there was a higher percentage of CI-positive neurons without LP compared to CI-negative neurons (CI-positive: 125/176, 71%; CI-negative: 8/18, 44%; *p* = 0.021). All data from neuronal counts, and the 2 × 3 crosstabulation and analyses results from both observers are provided in Appendix A and Table 2, respectively. Representative examples of the different CI and α-syn states are shown in Figure 2.

### 3.3. CIV Deficiency Is Not Associated with α-syn Pathology in Dopaminergic Neurons

A total of 167 (observer 1) and 141 (observer 2) single dopaminergic neurons from the SNc were evaluated from six different individuals. CIV-negative neurons were observed in four to five of the six individuals, depending on the observer (Appendix A). There was excellent inter-observer agreement [29] in the total number of observed neurons (ICC coefficient = 0.77, 95% C.I. (−0.073, 0.97), *p* < 0.001), CIV states (κ = 0.80, 95% C.I. (0.64, 0.95), *p* < 0.001), and α-syn states (κ = 0.92, 95% C.I. (0.85, 1.00), *p* < 0.001). As above, observer-1 results are further elaborated. Similar to the observation made in CI-negative neurons, we did not detect any LBs or PBs in CIV-negative neurons. Unlike the CI findings, however, there was no difference in the distribution of LP between CIV-positive and -negative neurons (Fisher–Freeman–Halton Exact Test, *p* = 0.44). Representative examples of the different CIV and α-syn states are shown in Figure 3. The 2 × 3 crosstabulation and results of the statistical analyses are shown in Table 3.

## 4. Discussion

We show that punctate α-syn aggregates, consistent with early stages of LP, show a strong predilection for CI-deficient dopaminergic SNc neurons. In contrast, late stages of LP (e.g., PBs and LBs) are found more frequently in neurons with intact CI immunostaining, in line with previous reports by us and others [13,20]. Our findings indicate that early α-syn aggregation is associated with CI deficiency in the dopaminergic SNc of individuals with iPD.

In our analyses, we identified early and late α-syn aggregation stages based on staining morphology. While these stages are based on the currently dominant view of LP evolution [6], it should be stressed that most of the supporting evidence comes from in vitro, cell and animal model studies, and the process of LP formation in the human brain is still not fully understood. Nevertheless, based on current knowledge [6], is likely that the observed punctate inclusions in our study do indeed represent early stages of LP.

Being observational, our data do not shed light on the nature or directionality of the relationship between α-syn aggregation and CI deficiency. One possibility is that α-syn aggregation causes CI deficiency in the SNc of individuals with iPD. This scenario is supported by multiple experiments in cell culture, showing that aggregated α-syn forms can localize to the inner mitochondrial membrane, inhibit CI function, and lead to bioenergetic impairment and increased ROS production [12,13,14,15,16]. Whether these observations are reflective of the events taking place in the brain of individuals with iPD remains, however, unknown.

Alternatively, CI deficiency may cause or promote α-syn aggregation. CI deficiency decreases the efficiency of oxidative phosphorylation, leading to ATP depletion [30]. This, in turn, could promote α-syn aggregation by impairing lysosomal and proteasomal function, both of which are ATP-dependent [31,32,33]. Moreover, CI deficiency is associated with increased production of ROS [30], which has been shown to exacerbate α-syn aggregation in cellulo and in vivo [34,35]. Treatment of rodents with the CI inhibitor rotenone causes dopaminergic degeneration of the SNc with formation of LB-like inclusions bearing high similarity to human LBs [36,37]. Moreover, based on a recent report, LP may be more prevalent in elderly individuals with mitochondrial diseases, mainly POLG-related disease and mitochondrial encephalomyopathy, lactic acidosis, and stroke-like episodes (MELAS) [38], both of which are characterized by extensive neuronal CI deficiency [39,40].

Yet another scenario is that α-syn aggregation and CI deficiency may be confounded by a common upstream etiology. Failure of autophagy and proteostasis due to lysosomal and proteasomal dysfunction of independent etiology, could, in theory, result in both mitochondrial dysfunction (due to impaired mitophagy) and α-syn aggregation.

Irrespective of the exact mechanisms linking α-syn aggregation and CI deficiency, the relative paucity of CI-negative neurons harboring mature LP, despite their strong predilection for early LP, suggests a double-hit mechanism, where neurons affected by both these pathology states are selectively lost during the disease. This is hardly unexpected, given that each of these pathologies, i.e., either α-syn aggregation or CI deficiency, is sufficient to cause dopaminergic degeneration in humans and other animals [41,42,43,44]. Moreover, a synergistic interaction between α-syn aggregation and CI deficiency in driving neurodegeneration in iPD is supported by animal studies. Mice overexpressing α-syn are more susceptible to neurodegeneration upon chemical CI inhibition [17], whereas α-syn knockout confers resistance to CI inhibition [18].

In agreement with previous findings by Reeve et al. [13], we found no association between CIV deficiency and either early or late LP stages. It must, however, be stressed that, due to the low number of observed CIV-negative neurons in our sample, there is a substantial risk for type II error in this analysis. Thus, an association between CIV and LP cannot be excluded based on our data.

In conclusion, our findings support a link between α-syn pathology and CI deficiency in the dopaminergic SNc of individuals with iPD. Specifically, our data suggest that early forms of LP have a strong predilection for CI deficient neurons. Furthermore, the combination of these two pathology states may drive neuronal loss, since late-stage LP appears to follow a reverse pattern, preferentially occurring in CI intact neurons. Future research should focus on addressing the dynamics and potential causality of the relationship between early α-syn aggregation and CI deficiency.

## Figures and Tables

**Figure 1 biomolecules-12-00747-f001:**
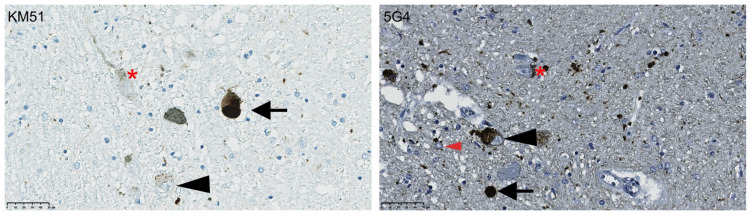
Immunostaining of the SNc with antibodies against full-length α-synuclein (KM51) or aggregated α-synuclein (5G4). Representative images of immunostaining against full length synuclein (KM51) and aggregated synuclein (5G4) in sections from the SNc of the same individual. Arrowheads show punctate inclusions, arrows show cytoplasmic and extracellular Lewy bodies and dense bodies. There is a higher level of immunopositive grains and threads, as well as inclusions in glial cells (recognized based on morphology; red arrowhead) in tissue stained with the 5G4 antibody. Neuromelanin is marked with an asterisk. Scalebar: 50 μm.

**Figure 2 biomolecules-12-00747-f002:**
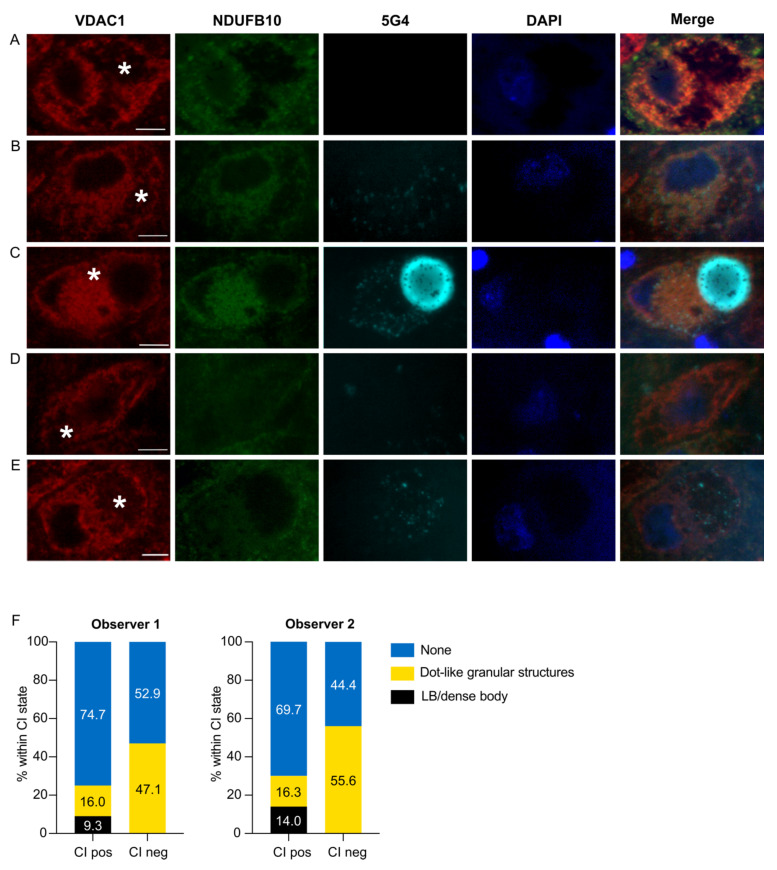
Immunofluorescence staining for CI and aggregated α-synuclein in dopaminergic neurons of the SNc. The images show representative examples of fluorescence immunoreactivity against the mitochondrial outer membrane marker (VDAC1, red), CI (NDUFB10, green), aggregated α-syn (5G4, cyan), dsDNA (DAPI, blue), and a combination of all channels. CI-positive neurons were either 5G4-negative (**A**), or 5G4-positive with punctate inclusions (**B**) or LB/PB (**C**). CI-negative neurons were either 5G4-negative (**D**), or positive with punctate inclusions (**E**). Neuromelanin is marked with an asterisk. Scale bar: 10 microns. The plots in (**F**) depict the total percentage of neurons in each α-syn inclusion state (LB or PB/punctate inclusions/no inclusion) within each CI state (positive/negative) from each of the observers.

**Figure 3 biomolecules-12-00747-f003:**
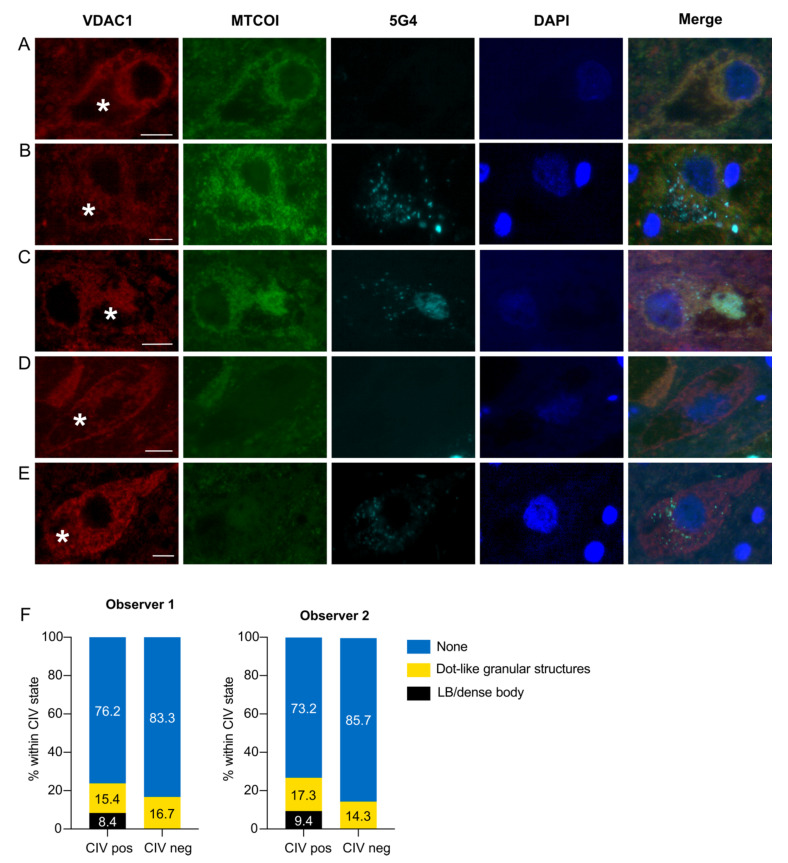
Immunofluorescence staining for CIV and aggregated α-synuclein in dopaminergic neurons of the SNc. Images show representative illustrations of fluorescence immunoreactivity against the mitochondrial outer membrane (VDAC1, red), CIV (MTCOI, green), aggregated α-syn (5G4, cyan), dsDNA (DAPI, blue), and a combination of all channels (merge). CIV-positive neurons were 5G4-negative (**A**), or positive with punctate inclusions (**B**) or LB/PB (**C**). CIV-negative neurons were either 5G4-negative (**D**), or positive with punctate inclusions (**E**). Neuromelanin is marked with an asterisk. Scale bar: 10 microns. The graphs in (**F**) depict the total percentage of neurons in each α-syn inclusion state (LB or PB/punctate inclusions/no inclusion) within each CIV state (positive/negative) from each of the observers.

**Table 1 biomolecules-12-00747-t001:** Demographic data of included cases.

Study ID	Diagnosis	AOD	DD	Sex	PMI	BRAAK Syn	Braak Tau	Thal Phase
PD 01	PD	88	11	M	30	6	III	1
PD 02	PD	82	12	M	46	6	I	2
PD 03	PD	78	12	F	24	6	III	3
PD 04	PD	90	10	F	24	6	III	0
PD 05	PD	72	10	F	72	6	II	0
PD 06	PD	75	12	M	11	6	I	0
PD 07	PD	94	14	M	36	6	IV	4
PD 08	PD	85	14	F	23	6	III	N/A
Ctrl 01	Ctrl	79	-	M	48	0	I	2
Ctrl 02	Ctrl	96	-	M	24	0	II	3
Ctrl 03	Ctrl	78	-	F	43	0	III	4
Ctrl 04	Ctrl	95	-	F	30	0	III	4
Ctrl 05	Ctrl	95	-	F	30	0	III	3

Diagnosis: pathological and clinical diagnosis; AOD: age of death; DD: disease duration in years; M: male; F: female; PMI: post-mortem interval; N/A: not available.

**Table 2 biomolecules-12-00747-t002:** A 2 × 3 contingency table of CI state (negative/positive) by α-syn inclusion (LB or PB/punctate inclusions/no inclusion).

			Cytoplasmic α-syn Inclusion						Total	
			LB/PB		Punctate		No Inclusion			
			O1	O2	O1	O2	O1	O2	O1	O2
**CI state**	**CI+**	Count	23	18	28	31	125	146	176	195
		Expected count	20.9	16.6	34.5	35.9	120.7	142.6		
		% Within CI state	14.0	9.2	15.9	15.9	71.0	74.9		
		Adjusted residual	1.6	1.3	−4.0	−3.2	2.3	2.0		
	**CI−**	Count	0	0	10	8	8	9	18	17
		Expected count	2.1	1.4	3.5	3.1	12.3	12.4		
		% Within CI state	0.0	0.0	55.6	47.1	44.4	52.9		
		Adjusted residual	−1.6	−1.3	4.0	3.2	−2.3	−2.0		
		*p*-value	0.11	0.19	6.3 × 10^−5^	1.3 × 10^−3^	0.021	0.046		
Total			23	18	38	39	133	155	194	212

α-syn: α-synuclein; LB: Lewy body; PB: pale body; Punctate: punctate inclusions; O1: observer 1; O2: observer 2; CI+: CI-positive neuron; CI−: CI-negative neuron.

**Table 3 biomolecules-12-00747-t003:** A 2 × 3 contingency table of CIV state (negative/positive) by α-syn inclusion (LB or PB/punctate inclusions/no inclusion).

			Cytoplasmic α-syn Inclusion						Total	
			LB/PB		Punctate		No Inclusion			
			O1	O2	O1	O2	O1	O2	O1	O2
**CIV state**	**CIV+**	Count	12	12	22	22	109	93	143	127
		Expected count	10.3	10.8	22.3	21.6	110.5	94.6		
		% Within CIV state	8.4	9.4	15.4	17.3	76.2	73.2		
		Adjusted residual	1.5	1.2	−0.2	0.3	−0.8	−1.0		
	**CIV−**	Count	0	0	4	2	20	12	24	14
		Expected count	1.7	1.2	3.7	2.4	18.5	10.4		
		% Within CIV state	0.0	0.0	16.7	14.3	83.3	85.7		
		Adjusted residual	−1.5	−1.2	0.2	−0.3	0.8	1.0		
		*p*-value	0.13	0.23	0.84	0.76	0.42	0.32		
Total			12	12	26	24	129	105	167	141

α-syn: α-synuclein; LB: Lewy body; PB: pale body; Punctate: punctate inclusions; O1: observer 1; O2: observer 2; CIV+: CIV-positive neuron; CIV−: CIV-negative neuron.

## Data Availability

The data supporting this study are available in the manuscript and Appendix A. All raw immunofluorescence images used for the assessments described in the study can be made available upon request to the corresponding author.

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
