# Peer review of "Early Forms of α-Synuclein Pathology Are Associated with Neuronal Complex I Deficiency in the Substantia Nigra of Individuals with Parkinson’s Disease"

_biomolecules, 2022, doi:10.3390/biom12060747_

Round 1
Reviewer 1 Report
In the present paper Flones and coauthors described that dopaminergic neurons with early asyn pathology are deficient of complex I, while neurons with LBs show intact complex I in patients with iPD.
The originality of the manuscript is poor and some observations are not enough objective and conclusive. In support of that, the discussion is mainly a description of the current literature.
Major revisions are required:
- Line 62: The sentence “The aggregation of LP” does not make sense. Alpha-synuclein is aggregated and not the pathology.
- Papers reporting mitochondrial complex I deficit in PD should be also reported.
- Line 93: “In line with previous findings” are not cited.
- Acquisition and analysis of immunofluorescence staining are not reported in the materials and method section.
- In Figure 1 it is stated that neuromelanin is marked with an asterisk but no asterisks are present in the figure.
- Staining of controls should be also reported.
- The LB are reported to be outside the neurons, for this reason it is not clear how the authors classified LBs in CI-positive neurons.
- Scale bar is missing from Figure 2 and 3.
- Figure 2 is not convincing. It is not clear whether the images want to represent dopaminergic neurons. Most notably, nuclei are almost not present within the cells. Assessed the VDAC1 staining, quadruple immunofluoscence for tyrosine hydroxylase could be more helpful to identify dopaminergic neurons and for the comprehension of the text.
Author Response
In the present paper Flones and coauthors described that dopaminergic neurons with early asyn pathology are deficient of complex I, while neurons with LBs show intact complex I in patients with iPD.
The originality of the manuscript is poor and some observations are not enough objective and conclusive. In support of that, the discussion is mainly a description of the current literature.
We thank the reviewer for the assessment of our manuscript. On a general level, we respectfully disagree with the Reviewer regarding lack of novelty in our study. Specifically, we would like to point out that, while the inverse association between LB and complex I deficiency is known - the observed positive association between early a-syn pathology and complex I loss is novel.
We addressing the Reviewer’s comments below.
Major revisions are required:
- Line 62: The sentence “The aggregation of LP” does not make sense. Alpha-synuclein is aggregated and not the pathology.
We agree with the Reviewer and have changed the sentence accordingly.
- Papers reporting mitochondrial complex I deficit in PD should be also reported.
There is a vast body of literature reporting CI deficiency in PD. In that regard, and in order to avoid selecting specific papers over others, we have chosen to cite a recent review paper (reference no 11) instead, which does an excellent job summarizing the current body of literature on the matter. We prefer this more fair approach over favorizing specific papers. That said, we will be happy to add more references at the discretion of the Editor.
- Line 93: “In line with previous findings” are not cited.
We regret this omission and have added the appropriate references.
- Acquisition and analysis of immunofluorescence staining are not reported in the materials and method section.
We have elaborated more on the methodology employed for the acquisition and analysis of the immunofluorescence staining – see lines 165-180. We will be happy to report additional specific details deemed necessary by the Reviewer and/or Editor.
- In Figure 1 it is stated that neuromelanin is marked with an asterisk but no asterisks are present in the figure.
The asterisks were not visible in the figure due to poor contrast. We have amended this and updated the figure.
- Staining of controls should be also reported.
Controls were negative for Lewy pathology. This is reported in the Results, lines 201-202. Since controls did not harbor Lewy pathology, it would not be meaningful to include them in the immunofluorescence experiments and downstream analyses.
- The LB are reported to be outside the neurons, for this reason it is not clear how the authors classified LBs in CI-positive neurons.
Lewy pathology arises and matures inside cells. Therefore, LBs are primarily and mainly an intracellular (mostly intraneuronal) phenomenon. Extracellular LBs are believed to remain after the death and breakdown of the hosting neuron. The extracellular LB shown in Fig-1 is only an indicative example of such a case. Extracellular LBs were not included in the immunofluorescent analyses assessing the relationship between LP and CI or CIV states. Only intraneuronal LP was used in these analyses, as depicted in Fig 2 and 3.
- Scale bar is missing from Figure 2 and 3.
This has been amended in the figures.
- Figure 2 is not convincing. It is not clear whether the images want to represent dopaminergic neurons. Most notably, nuclei are almost not present within the cells. Assessed the VDAC1 staining, quadruple immunofluoscence for tyrosine hydroxylase could be more helpful to identify dopaminergic neurons and for the comprehension of the text.
Nuclei are present in all dopaminergic neurons depicted. However, the nuclei of dopaminergic neurons are typically large and rich in euchromatin, which stains much weaker by DAPI compared to heterochromatin. For this reason, the nuclei of dopaminergic neurons are typically weakly stained by DAPI compared to those of surrounding glia and other neuronal types. That said, we have increased the contrast of the DAPI images used in Fig 2 and 3, rendering the nuclei of the dopaminergic neurons more visible.
In terms of recognizing the dopaminergic neurons, this was done on the basis of neuronal morphology, size and cytoplasmic neuromelanin content. These features allowed us to identify the dopaminergic neurons with high confidence. We have indicated this in the figures and added asterisks to indicate the presence of the neuromelanin. While we agree with the Reviewer that TH-staining would be helpful, this is not possible to implement in this case because we are limited to four channels. To add TH, we would have to sacrifice one of the four channels (CI or CIV, a-syn, VDAC, and DAPI), all of which are essential to our analyses. Removing VDAC1 would invalidate the CI or CIV assessment as we would not know whether neurons have a specific complex deficiency or simply low mitochondrial mass (which may occur for biological or technical reasons). Removing DAPI would not allow us to ensure that we only assess neurons with visible nuclei of intact morphology (i.e., neurons that were alive/viable at the time of death).
Given these technical limitations, we believe that we have chosen the most informative target combination for the purpose of these experiments.
Reviewer 2 Report
Overall, the manuscript is well written and the experiments seem to be carefully performed.
I have 2 main concerns:
- How do you know that some of the cells containing aggregated alpha-synuclein in Figure 1 (5G4 antibody) are glial cells? Meaning, how do you distinguish/identify glial cells?
- Although mitochondrial complex I dysfunction/function deficiency has been related with PD, it is a little weird to me that there are some neurons that do not express complex I at all. Meaning, one thing is that complex I in mitochondria does not function properly and another one is that its expression is completely lost... Any explanation?
Author Response
Overall, the manuscript is well written and the experiments seem to be carefully performed.
We thank the reviewer for the positive assessment of our manuscript and are addressing the comments below.
I have 2 main concerns:
How do you know that some of the cells containing aggregated alpha-synuclein in Figure 1 (5G4 antibody) are glial cells? Meaning, how do you distinguish/identify glial cells?
Glia were identified based on morphology, including size, nuclear density and shape. The identification of glial LP using the 5G4 antibody has been previously described (PMID: 22370907).
Although mitochondrial complex I dysfunction/function deficiency has been related with PD, it is a little weird to me that there are some neurons that do not express complex I at all. Meaning, one thing is that complex I in mitochondria does not function properly and another one is that its expression is completely lost... Any explanation?
Complex I negative neurons, defined as neurons showing no visible/detectable immunostaining for complex I (in spite of being positive for a mitochondrial mass marker like VDAC1) is a common phenomenon in PD, which is encountered both in the substantia nigra and numerous other brain regions. This has been reported in numerous studies by us and others (PMID: 1665052, 29270838, 33745923).
A similar but more widespread phenomenon (i.e., affecting a higher proportion of neurons) is seen in the brain of patients with various forms of mitochondrial disease (PMID: 24841123).
Negative immunostaining in this regard does not necessarily mean that complex I expression is completely lost, but it must be severely reduced to levels below the detection threshold of immunostaining. Notably, negative neurons are found also when immunostaining techniques with very high sensitivity are employed, involving signal amplification by polymerization (PMID: 29270838).
Round 2
Reviewer 1 Report
Authors appropriately responded to the reviewer's comments